# Can Unconditional Language Models Recover Arbitrary Sentences?

**Nishant Subramani**
New York University
nishant@nyu.edu

**Samuel R. Bowman**
New York University

**Kyunghyun Cho**
New York Univeristy
Facebook AI Research
CIFAR Azrieli Global Scholar

## Abstract

Neural network-based generative language models like ELMo and BERT can work effectively as general purpose sentence encoders in text classification without further fine-tuning. Is it possible to adapt them in a similar way for use as general-purpose decoders? For this to be possible, it would need to be the case that for any target sentence of interest, there is some continuous representation that can be passed to the language model to cause it to reproduce that sentence. We set aside the difficult problem of designing an encoder that can produce such representations and, instead, ask directly whether such representations exist at all. To do this, we introduce a pair of effective, complementary methods for feeding representations into pretrained unconditional language models and a corresponding set of methods to map sentences into and out of this representation space, the *reparametrized sentence space*. We then investigate the conditions under which a language model can be made to generate a sentence through the identification of a point in such a space and find that it is possible to recover arbitrary sentences nearly perfectly with language models and representations of moderate size *without modifying any model parameters*.

## 1 Introduction

We have recently seen great successes in using pretrained language models as encoders for a range of difficult natural language processing tasks (Dai and Le, 2015; Peters et al., 2017, 2018; Radford et al., 2018; Ruder and Howard, 2018; Devlin et al., 2018; Dong et al., 2019; Yang et al., 2019), often with little or no fine-tuning: Language models learn useful representations that allow them to serve as *general-purpose encoders*. A hypothetical *general-purpose decoder* would offer similar benefits: making it possible to both train models for text generation tasks with little annotated data and share parameters extensively across applications in environments where memory is limited. Then, is it possible to use a pretrained language model as a *general-purpose decoder* in a similar fashion?

For this to be possible, we would need both a way of feeding some form of continuous sentence representation into a trained language model and a task-specific encoder that could convert some task input into a sentence representation that would cause the language model to produce the desired sentence. We are not aware of any work that has successfully produced an encoder that can interoperate in this way with a pretrained language model, and in this paper, we ask whether it is possible at all: Are typical, trained neural network language models capable of recovering arbitrary sentences through conditioning of this kind?

We start by defining the *sentence space* of a recurrent language model and show how this model maps a given sentence to a trajectory in this space. We reparametrize this sentence space into a new space, the *reparametrized sentence space*, by mapping each trajectory in the original space to a point in the new space. To accomplish the reparametrization, we introduce two complementary methods to add

additional bias terms to the previous hidden and cell state at each time step in the trained and frozen language model, and optimize those bias terms to maximize the likelihood of the sentence.

Recoverability inevitably depends on model size and quality of the underlying language model, so we vary both along with different dimensions for the reparametrized sentence space. We find that the choice of optimizer (nonlinear conjugate gradient over stochastic gradient descent) and initialization are quite sensitive, so it is unlikely that a simple encoder setup would work out of the box.

Our experiments reveal that we can achieve full recoverability with a reparametrized sentence space with dimension equal to the dimension of the recurrent hidden state of the model, at least for large enough models: For nearly all sentences, there exists a single vector that can recover the sentence perfectly. We show that this trend holds even with sentences that come from a different domain than the ones used to train the fixed language model. We also observe that the smallest dimension able to achieve the greatest recoverability is approximately equal to the dimension of the recurrent hidden state of the model. Furthermore, we observe that recoverability decreases as sentence length increases and that models find it increasingly difficult to generate words later in a sentence. In other words, models rarely generate any correct words after generating an incorrect word when decoding a given sentence. Lastly, experiments on recovering random sequences of words show that our reparametrized sentence space does not simply memorize the sequence, but also utilizes the language model. These observations indicate that unconditional language models can indeed be conditioned to recover arbitrary sentences almost perfectly and may have a future as universal decoders.

## 2 The Sentence Space of a Recurrent Language Model

In this section, we first cover the background on recurrent language models. We then characterize its sentence space and show how we can reparametrize it for easier analysis. In this reparametrized sentence space, we define the recoverability of a sentence.

### 2.1 Recurrent Language Models

**Model Description** We train a 2-layer recurrent language model over sentences autoregressively:

$$p(x_1, \ldots, x_T) = \prod_{t=1}^{T} p(x_t | x_1, \ldots, x_{t-1}) \tag{1}$$

A neural network models each conditional distribution (right side) by taking as input all the previous tokens $(x_1, \ldots, x_{t-1})$ and producing as output the distribution over all possible next tokens. At every time-step, we update the internal hidden state $h_{t-2}$, which summarizes $(x_1, \ldots, x_{t-2})$, with a new token $x_{t-1}$, resulting in $h_{t-1}$. This resulting hidden state, $h_{t-1}$, is used to compute $p(x_t | x_1, \ldots, x_{t-1})$:

$$h_{t-1} = f_\theta(h_{t-2}, x_{t-1}), \tag{2}$$

$$p(x_t = i | x_1, \ldots, x_{t-1}) = g_\theta^i(h_{t-1}), \tag{3}$$

where $f_\theta : \mathbb{R}^d \times V \to \mathbb{R}^d$ is a recurrent transition function often implemented as an LSTM recurrent network (as in Hochreiter and Schmidhuber, 1997; Mikolov et al., 2010). The readout function $g$ is generally a softmax layer with dedicated parameters for each possible word. The incoming hidden state $h_0 \in \mathbb{R}^d$ at the start of generation is generally an arbitrary constant vector. We use zeroes. For a LSTM language model with $l$ layers of $d$ LSTM units, its model dimension $d^* = 2dl$ because LSTMs have two hidden state vectors (conventionally h and c) both of dimension $d$.

**Training** We train the full model using stochastic gradient decent with negative log likelihood loss.

**Inference** Once learning completes, a language model can be straightforwardly used in two ways: scoring and generation. To score, we compute the log-probability of a newly observed sentence according to Eq. (1). To generate, we use ancestral sampling by sampling tokens $(x_1, \ldots, x_T)$ sequentially, conditioning on all previous tokens at each step via Eq. (1).

In addition, we can find the *approximate* most likely sequence using beam search (Graves, 2012). This procedure is generally used with language model variants like sequence-to-sequence models (Sutskever et al., 2014) that condition on additional *context*. We use this procedure in backward estimation to recover the sentence corresponding to a given point in the reparametrized space.

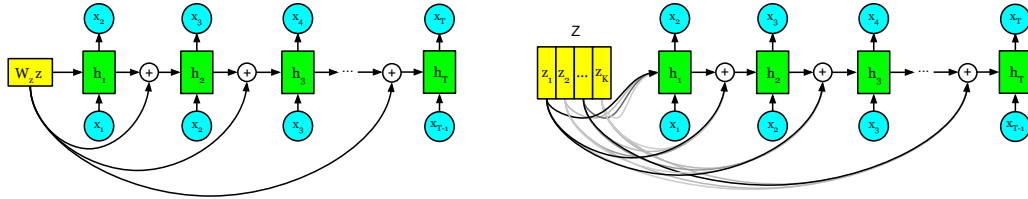

Figure 1: We add an additional bias, $W_z z$ (left, when $\dim(z) \leq d^*$) or $Z = [z_1 \ldots z_K]$ (right, when $dim(z) > d^*$), to the previous hidden and cell state at every time step. Only the $z$ vector or $Z$ matrix is trained during forward estimation: The main LSTM parameters are frozen and $W_z$ is set randomly. In the right hand case, we use soft attention to allow the model to use different slices of $Z$ each step.

## 2.2 Defining the Sentence Space

The recurrent transition function $f_\theta$ in Eq. (2) defines a dynamical system driven by the observations of tokens $(x_1, \ldots, x_T) \in \mathcal{X}$ in a sentence. In this dynamical system, all trajectories start at the origin $h_0 = [0, \ldots, 0]^\top$ and evolve according to incoming tokens ($x_t$'s) over time. Any trajectory $(h_0, \ldots, h_T)$ is entirely embedded in a $d$-dimensional space, where $d$ is equal to the dimension of the hidden state and $\mathcal{H} \in \mathbb{R}^d$, i.e., $h_t \in \mathcal{H}$. In other words, the language model embeds a sentence of length $T$ as a $T + 1$-step trajectory in a $d$-dimensional space $\mathcal{H}$, which we refer to as the *sentence space* of a language model.

**Reparametrizing the Sentence Space** We want to recover sentences from semantic representations that do not encode sentence length symbolically. Given that and since a single replacement of an intermediate token can drastically change the remaining trajectory in the sentence space, we want a flat-vector representation. In order to address this, we propose to (approximately) reparametrize the sentence space into a flat-vector space $\mathcal{Z} \in \mathbb{R}^{d'}$ to characterize the sentence space of a language model. Under the proposed reparameterization, a trajectory of hidden states in the sentence space $\mathcal{H}$ maps to a vector of dimension $d'$ in the reparametrized sentence space $\mathcal{Z}$. To accomplish this, we add bias terms to the previous hidden and cell state at each time step in the model and optimize them to maximize the log probability of the sentence as shown in Figure 1. We add this bias in two ways: (1) if $d' \leq d^*$, we use a random projection matrix to project our vector $z \in \mathbb{R}^{d'}$ up to $d^*$ and (2) if $d' > d^*$, we use soft-attention with the previous hidden state to adaptively project our vector $z \in \mathbb{R}^{d'}$ down to $d^*$ (Bahdanau et al., 2015).

Our reparametrization must approximately allow us to go back (forward estimation) and forth (backward estimation) between a sequence of tokens, $(x_1, \ldots, x_T)$, and a point $z$ in this reparametrized space $\mathcal{Z}$ *via* the language model. We need back-and-forth reparametrization to measure recoverability. Once this back-and-forth property is established, we can inspect a set of points in $\mathcal{Z}$ instead of trajectories in $\mathcal{H}$. A vector $z \in \mathcal{Z}$ resembles the output of an encoder acting as context for a conditional generation task. This makes analysis in $\mathcal{Z}$ resemble analyses of context on sequence models and thus helps us understand the unconditional language model that we are trying to condition with $z$ better.

We expect that our reparametrization will allow us to approximately go back and forth between a sequence and its corresponding point $z \in \mathcal{Z}$ because we expect $z$ to contain all of the information of the sequence. Since we're adding $z$ at every time-step, the information preserved in $z$ will not degrade as quickly as the sequence is processed like it could if we just added it to the initial hidden and cell states. While there are other similar ways to integrate $z$, we choose to modify the recurrent connection.

**Using the Sentence Space** In this paper, we describe the reparametrized sentence space $\mathcal{Z}$ of a language model as a set of $d'$-dimensional vectors that correspond to a set $D'$ of sentences that were not used in training the underlying language model. This use of unseen sentences helps us understand the sentence space of a language model in terms of generalization rather than memorization, providing insight into the potential of using a pretrained language model as a fixed decoder/generator. Using our reparametrized sentence space framework, evaluation techniques designed for investigating word vectors become applicable. One of those interesting techniques that we can do now is interpolation between different sentences in our reparameterized sentence space (Table 1 in Choi et al., 2017; Bowman et al., 2016), but we do not explore this here.

**Forward Estimation** $\mathcal{X} \to \mathcal{Z}$ The goal of forward estimation is to find a point $z \in \mathcal{Z}$ that represents a sentence $(x_1, \ldots, x_T) \in \mathcal{X}$ via the trained language model (i.e., fixed $\theta$). When the dimension of $z$ is smaller than the model dimension $d^*$, we use a random projection matrix to project it up to $d^*$ and when the dimension of $z$ is greater than the model dimension, we use soft attention to project it down to $d^*$. We modify the recurrent dynamics $f_\theta$ in Eq. (2) to be:

$$h_{t-1} = f_\theta(h_{t-2} + \underline{z}', x_{t-1}) \tag{4}$$

$$z' = \begin{cases} W_z z, & \text{if } \dim(z) \leq d^* \\ \text{softmax}(h_{t-2}^\top Z)Z^\top, & \text{if } \dim(z) > d^* \end{cases} \tag{5}$$

where $Z \in \mathbb{R}^{d \times k}$ and is just the unflattened matrix of $z$ consisting of $k = \dim(z)/d$ vectors of dimension $d$. We initialize the hidden state by $h_0 = \underline{z}'$. $W_z \in \mathbb{R}^{d \times d'}$ is a random matrix with $L_2$-normalized rows, following Li et al. (2018) and is an identity matrix when $d = d'$: $W_z = [w_z^1; \ldots; w_z^{d'}]$, where $w_z^l = \epsilon^l / \|\epsilon^l\|_2$ and $\epsilon^l \sim \mathcal{N}(0, 1^2) \in \mathbb{R}^d$. We then estimate $z$ by maximizing the log-probability of the given sentence under this modified model, while fixing the original parameters $\theta$:

$$\hat{z} = \operatorname*{argmax}_{z \in \mathcal{Z}} \sum_{t=1}^{T} \log p(x_t | x_{<t}, z) \tag{6}$$

We represent the entire sentence $(x_1, \ldots, x_T)$ in a single $z$. To solve this optimization problem, we can use any off-the-shelf gradient-based optimization algorithm, such as gradient descent or nonlinear conjugate descent. This objective function is highly non-convex, potentially leading to multiple approximately optimal $z$'s. As a result, to estimate $z$ in forward estimation, we use nonlinear conjugate gradient (Wright and Nocedal, 1999) implemented in SciPy (Jones et al., 2014) with a limit of 10,000 iterations, although almost all runs converge much more quickly. Our experiments, however, reveal that many of these $z$'s lead to similar performance in recovering the original sentence.

**Backward Estimation** $\mathcal{Z} \to \mathcal{X}$ Backward estimation, an instance of sequence decoding, aims at recovering the original sentence $(x_1, \ldots, x_T)$ given a point $z$ in the reparametrized sentence $\mathcal{Z}$, which we refer to as *recovery*. We use the same objective function as in Eq. (6), but we optimize over $(x_1, \ldots, x_T)$ rather than over $z$. Unlike forward estimation, backward estimation is a combinatorial optimization problem and cannot be solved easily with a recurrent language model (Cho, 2016; Chen et al., 2018). To circumvent this, we use beam search, which is a standard approach in conditional language modeling applications such as machine translation. Our backward estimation procedure does not assume a true length when decoding with beam search—we stop when an end of token or 100 tokens is reached.

## 2.3 Analyzing the Sentence Space through Recoverability

Under this formulation, we can investigate various properties of the sentence space of the underlying model. As a first step toward understanding the sentence space of a language model, we propose three round-trip recoverability metrics and describe how we use them to characterize the sentence space.

**Recoverability** *Recoverability* measures how much information about the original sentence $x = (x_1, \ldots, x_T) \in \mathcal{X}$ is preserved in the reparameterized sentence space $\mathcal{Z}$. We measure this by reconstructing the original sentence $x$. First, we forward-estimate the sentence vector $z \in \mathcal{Z}$ from $x \in \mathcal{X}$ by Eq. (6). Then, we reconstruct the sentence $\hat{x}$ from the estimated $z$ via backward estimation. To evaluate the quality of reconstruction, we compare the original and reconstructed sentences, $x$ and $\hat{x}$ using the following three metrics:

1. Exact Match (EM): $\sum_{t=1}^{T} \mathbb{I}(x_t = \hat{x}_t)/T$
2. **BLEU** (Papineni et al., 2002)
3. Prefix Match (PM): $\operatorname*{argmax}_t \text{EM}(x_{\leq t} = \hat{x}_{\leq t})/T$

Exact match gives information about the possibility of perfect recoverability. BLEU provides us with a smoother approximation to this, in which the hypothesis gets some reward for n-gram overlap, even if slightly inexact. Since BLEU is 0 for sentences with less than 4 tokens, we smooth these by only considering n-grams up to the sentence length if sentence length is less than 4. Prefix match measures the longest consecutive sequence of tokens that are perfectly recovered from the beginning of the

sentence and we divide this by the sentence length. We use prefix match because early experiments show a very strong left-to-right falloff in quality of generation. In other words, candidate generations are better for shorter sentences and once an incorrect token is generated, future tokens are extremely unlikely to be correct. We compute each metric for each sentence $x \in D'$ by averaging over multiple optimization runs, we show exact match (EM) in the equations, but we do the same for BLEU and Prefix Match. To counter the effect of non-convex optimization in Eq. (6), these runs vary by the initialization of $z$ and the random projection matrix $W_z$ in Eq. (4). That is,

$$\overline{\text{EM}}(x, \theta) = \mathbb{E}_{z_0 \in \mathcal{Z}} \left[ \mathbb{E}_{W_z \in \mathbb{R}^{d \times d'}} \left[ \text{EM}(x, \hat{x}) \right] \right]$$

**Effective Dimension by Recoverability** These recoverability measures allow us to investigate the underlying properties of the proposed sentence space of a language model. If all sentences can be projected into a $d$-dimensional sentence space $\mathcal{Z}$ and recovered perfectly, the effective dimension of $\mathcal{Z}$ must be no greater than $d$. In this paper, when analyzing the effective dimension of a sentence space of a language model, we focus on the effective dimension given a target recoverability $\tau$:

$$\hat{d}'(\theta, \tau) = \min \left\{ d' \left| \overline{\text{EM}}(D', \theta) > \tau \right. \right\} \tag{7}$$

where $\overline{\text{EM}}(D', \theta) = \frac{1}{|D'|} \sum_{x \in D'} \overline{\text{EM}}(x, \theta)$. In other words, given a trained model ($\theta$), we find the smallest effective dimension $d'$ (the dimension of $\mathcal{Z}$) that satisfies the target recoverability ($\tau$). Using this, we can answer questions like what is the minimum dimension $d'$ needed to achieve recoverability $\tau$ under the model $\theta$. Using this, the unconstrained effective dimension, i.e. the smallest dimension that satisfies the best possible recoverability, is:

$$\hat{d}'(\theta) = \underset{d' \in \{1, \ldots, d\}}{\operatorname{argmin}} \max \frac{1}{|D'|} \sum_{x \in D'} \overline{\text{EM}}(x, \theta) \tag{8}$$

We approximate the effective dimension by inspecting various values of $d'$ on a logarithmic scale: $d' = 128, 256, 512, \ldots, 32768$. Since our forward estimation process uses non-convex optimization and our backward estimation process uses beam search, our effective dimension estimates are upper-bound approximations.

## 3 Experimental Setup

**Corpus** We use the fifth edition of the English Gigaword (Graff et al., 2003) news corpus. Our primary model is trained on 50M sentences from this corpus, and analysis experiments additionally include a weaker model trained on a subset of only 10M. Our training sentences are drawn from articles published before November 2010. We use a development set with 879k sentences from the articles published in November 2010 and a test set of 878k sentences from the articles published in December 2010. We lowercase the entire corpus, segment each article into sentences using NLTK (Bird and Loper, 2004), and tokenize each sentence using the Moses tokenizer (Koehn et al., 2007). We further segment the tokens using byte-pair encoding (BPE; following Sennrich et al., 2016) with 20,000 merges to obtain a vocabulary of 20,234 subword tokens. To evaluate out-of-domain sentence recoverability, we use a random sample of 50 sentences from the IWSLT16 English to German translation dataset (validation portion) processed in the same way and using the same vocabulary.

**Recurrent Language Models** The proposed framework is agnostic to the underlying architecture of a language model. We choose a 2-layer language model with LSTM units (Graves, 2013). We construct a small, medium, and large language model consisting of 256, 512, and 1024 LSTM units respectively in each layer. The input and output embedding matrices of 256, 512, and 1024-dimensional vectors respectively are shared (Press and Wolf, 2017). We use dropout (Srivastava et al., 2014) between the two recurrent layers and before the final linear layer with a drop rate of 0.1, 0.25, and 0.3 respectively. We use stochastic gradient descent with Adam with a learning rate of $10^{-4}$ on 100-sentence minibatches (Kingma and Ba, 2014), where sentences have a maximum length of 100.

We measure perplexity on the development set every 10k minibatches, halve the learning rate whenever it increases, and clip the norm of the gradient to 1 (Pascanu et al., 2013). For each training set (10M and 50M), we train for only one epoch. Because of the large size of the training sets, these models nonetheless achieve a good fit to the underlying distribution (Table 1).

Table 1: Language modeling perplexities on English Gigaword for the models under study.

| Model | $d$ | |Train| = 10M | | |Train| = 50M | |
|---|---|---|---|---|---|
| | | Dev Ppl. | Test Ppl. | Dev Ppl. | Test Ppl. |
| SMALL | 256 | 122.9 | 125.2 | 77.2 | 79.2 |
| MEDIUM | 512 | 89.6 | 91.3 | 62.1 | 63.5 |
| LARGE | 1024 | 65.9 | 67.7 | **47.4** | **48.9** |

**Reparametrized Sentence Spaces** We use a set $D'$ of 100 randomly selected sentences from the development set in our analysis. We set $z$ to have 128, 256, 512, 1024, 2048, 4096, 8192, 16384 and 32768 dimensions for each language model and measure its recoverability. For each sentence we have ten random initializations. When the dimension $d'$ of the reparametrized sentence space is smaller than the model dimension, we construct ten random projection matrices that are sampled once and fixed throughout the optimization procedure. We perform beam search with beam width 5.

# 4   Results and Analysis

**Recoverability Results** In Figure 2, we present the recoverability results of our experiments relative to sentence length using the three language models trained on 50M sentences. We observe that the recoverability increases as $d'$ increases until $d' = d^*$. After this point, recoverability plateaus. Recoverability between metrics for a single model are strongly positively correlated. We also observe that recoverability is nearly perfect for the large model when $d' = 4096$ achieving EM $\geq 99$, and very high for the medium model when $d' \geq 2048$ achieving EM $\geq 84$.

We find that recoverability increases for a specific $d'$ as the language model is trained, although we cannot present the result due to space constraints. The corresponding figure to Figure 2 for the 10M setting and tables for both of the settings detailing overall performance are provided in the appendix. All these estimates have high confidence (small standard deviations).

**Effective Dimension of the Sentence Space** From Figure 2, the large model's unconstrained effective dimension is $d^* = 4096$ with a slight degradation in recoverability when increasing $d'$ beyond $d^*$. For the medium model, we notice that its unconstrained effective dimension is also $d^* = 2048$ with no real recoverability improvements when increasing $d'$ beyond $d^*$. For the small model, however, its unconstrained effective dimension is 8192, which is much greater than $d^* = 1024$.

When $d' = 4096$, we can recover any sentence nearly perfectly, and for large sentences, the large model with $d' \geq 4096$ achieves recoverability estimates $\tau \geq 0.8$. For other model sizes and other dimensions of the reparametrized space, we fail to perfectly recover some sentences. To ascertain which sentences we fail to recover, we look at the shapes of each curve. We observe that the vast majority of these curves never increase, indicating recoverability and sentence length have a strong negative correlation. Most curves decrease to 0 as sentence length exceeds 30 indicating that longer sentences are more difficult to recover. Earlier observations in using neural sequence-to-sequence models for machine translation concluded exactly this (Cho et al., 2014; Koehn and Knowles, 2017).

This suggests that a fixed-length representation lacks the capacity to represent a complex sentence and could sacrifice important information in order to encode others. The degradation in recoverability also implies that the unconstrained effective dimension of the sentence space could be strongly related to the length of the sentence and may not be related to the model dimension $d^*$. The fact that the smaller model has an unconstrained effective dimension much larger than $d^*$ supports this claim.

**Impact of Beam Width & Optimization Strategy** To analyze the impact of various beam widths, we experimented with beam widths of 5, 10, and 20 in decoding. We find that results are consistent across these beam widths. As a result, all experimental results in this paper other than this one use a beam width of 5. We provide a representative partial table of sentence recoverability varying just beam width during decoding in Table 2.

To understand the importance of the choice of optimizer, we experimented with using Adam with a learning rate of $10^{-4}$ with default settings on our best performing settings for each model size. We find that using Adam results in recovery estimates that do not exceed 1.0 BLEU for all three situations, hinting at the highly non-convex nature of the optimization problem.

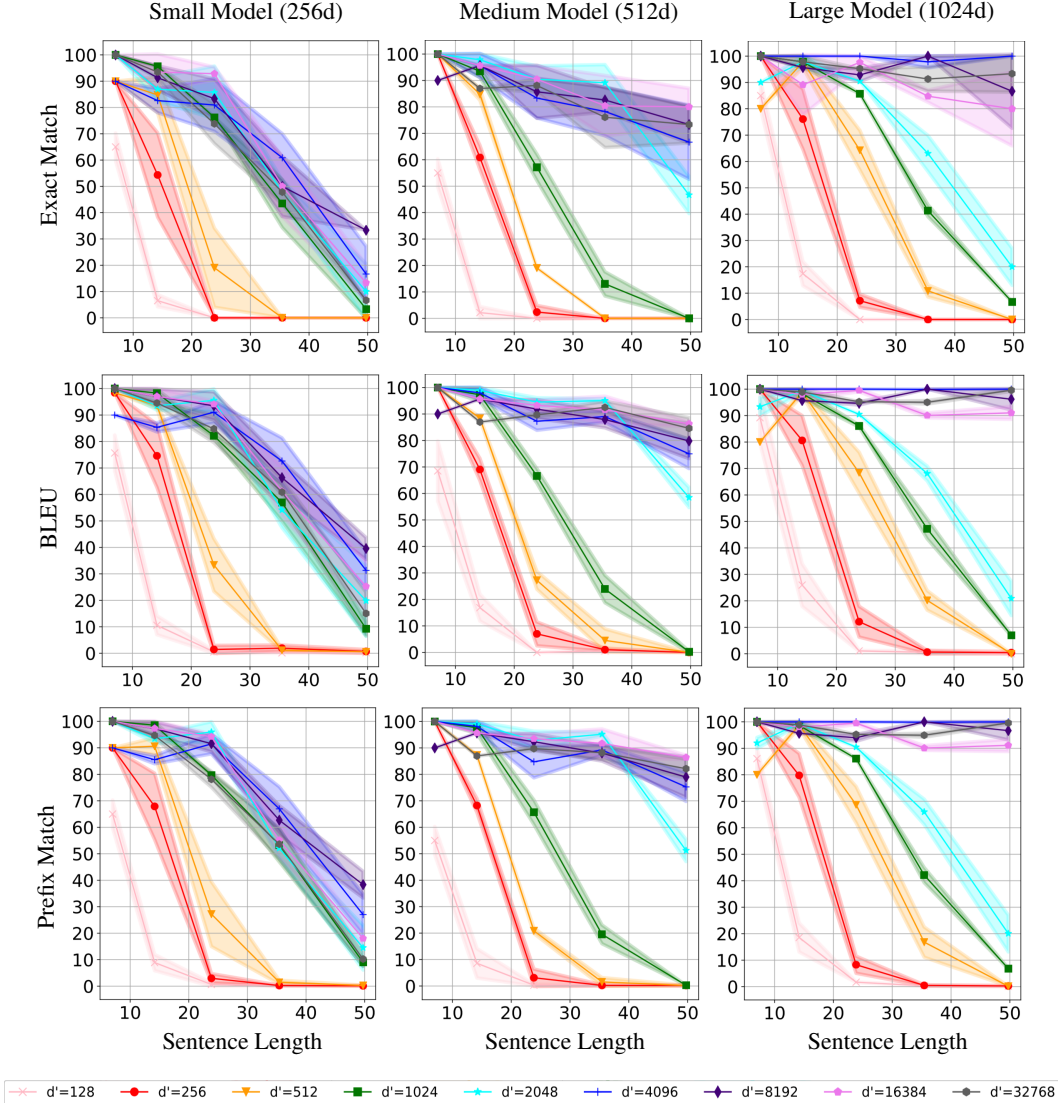

Figure 2: Plots of the three recoverability metrics with respect to varying sentence lengths for each of our three model sizes for the 50M sentence setting. Within each plot, the curves correspond to the varying dimensions of $z$ including error regions of $\pm\sigma$. Regardless of metric, recoverability improves as the size and quality of the language model and dimension of the reparametrized sentence space increases. The corresponding plot for the 10M sentence setting is in the appendix.

**Sources of Randomness** There are two points of stochasticity in the proposed framework: the non-convexity of the optimization procedure in forward estimation (Eq. 6) and the sampling of a random projection matrix $W_z$. However, based on the small standard deviations in Figure 2, these have minimal impact on recoverability. Also, the observation of high confidence (low-variance) upper-bound estimates for recoverability supports the usability of our recoverability metrics for investigating a language model's sentence space.

**Out-of-Domain Recoverability** To study how well our pretrained language models can recover sentences out-of-domain, we evaluate recoverability on our IWSLT data. IWSLT is comrpised of TED talk transcripts, a very different style than the news corpora our language models were trained on. The left and center graphs in Figure 3 show that recovery performance measured in BLEU is nearly perfect even for out-of-domain sentences for both the medium and large models when $d' \geq d^*$, following trends from the experiments on English Gigaword from Figure 2.

**More than just Memorization** Near-perfect performance on out-of-domain sentences indicates that this methodology could either be learning important properties of language by leveraging the language

Table 2: Recoverability (BLEU) varying beam width on English Gigaword

| Model | $|Z|$ | BLEU | | |
|---|---|---|---|---|
| | | Width=5 | Width=10 | Width=20 |
| SMALL; 50M | 512 | 40.0 | 40.3 | 40.5 |
| SMALL; 50M | 8192 | 81.1 | 79.8 | 79.6 |
| MEDIUM; 50M | 512 | 41.1 | 41.1 | 42.3 |
| MEDIUM; 50M | 16384 | 92.4 | 91.9 | 89.8 |
| LARGE; 50M | 512 | 54.8 | 54.1 | 53.8 |
| LARGE; 50M | 4096 | 99.8 | 99.8 | 99.5 |

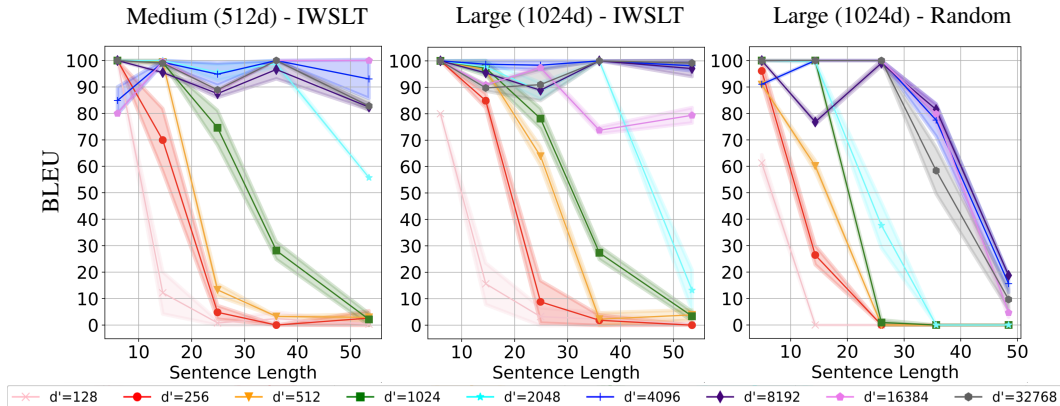

Figure 3: Recoverability (BLEU) on IWSLT for medium (left) and large models (center) and on the random data for the large model (right).

model, which helps generalization, or just be memorizing any arbitrary sequence without using the language model at all. To investigate this, we randomly sample 50 sentences of varying lengths where each token is sampled randomly with equal probability with replacement from the vocabulary. The right graph in Figure 3 shows BLEU recovery for the large model. Many of the shorter sequences can be recovered well, but for sequences greater than 25 subword units, recoverability drops quickly. This experiment shows that memorization cannot fully explain results on Gigaword or IWSLT16.

**Towards a General-Purpose Decoder** In this formulation, our vector $z'$ can be considered as trainable context used to condition our unconditioned language models to generate arbitrary sentences. Since we find that well-trained language models of reasonable size have an unconstrained effective dimension with high recoverability that is approximately its model dimension on both in-domain and out-of-domain sequences, unconditional language models are able to utilize our context $z'$ effectively. Further experiments confirm that our context vectors do not simply memorize arbitrary sequences, but leverage the language model to generate well-formed sequences. As a result, such a model could be used as a task-independent decoder given an encoder with the ability to generate an optimal context vector $z'$.

We observe that recoverability isn't perfect for both the small and medium models, falling off dramatically for longer sentences, indicating that the minimum model size for high recoverability is fairly large. Since the sentence length distribution is a Zipf distribution (heavily right-tailed), if we can increase the recoverability degredation cutoff point, the number of sentences we fail to recover perfectly would decrease exponentially. However, since we find that larger and better-trained models can exhibit near perfect recoverability for both in-domain and out-of-domain sequences and can more easily utilize our conditioning strategy, we think that this may only be a concern for lower capacity models. Our methodology could use a regularization mechanism to smooth the implicit sentence space. This may improve recoverability and reduce the unconstrained effective dimension, whereby increasing the applicability of an unconditional language model as a general-purpose decoder.

# 5   Related Work

**Latent Variable Recurrent Language Models** The way we describe the sentence space of a language model can be thought of as performing inference over an implicit latent variable $z$ using a fixed decoder $\theta$. This resembles prior work on sparse coding (Olshausen and Field, 1997) and generative latent optimization (Bojanowski et al., 2018). Under this lens, it also relates to work on training latent variable language models, such as models based on variational autoencoders by Bowman et al. (2016) and sequence generative adversarial networks by Yu et al. (2017). The goal of identifying the smallest dimension of the sentence space for a specific target recoverability resembles work looking at continuous bag-of-words representations by Mu et al. (2017). Our approach differs from these approaches in that we focus entirely on analyzing a fixed model that was trained unconditionally. Our formulation of the sentence space also is more general, and potentially applies to all of these models.

**Pretrained Recurrent Language Models** Pretrained or separately trained language models have largely been used in two contexts: as a feature extractor for downstream tasks and as a scoring function for a task-specific decoder (Gulcehre et al., 2015; Li et al., 2016; Sriram et al., 2018). None of the above analyze how a pretrained model represents sentences nor investigate the potential of using a language model as a decoder. The work by Zoph et al. (2016) transfers a pretrained language model, as a part of a neural machine translation system, to another language pair and fine-tunes. The positive result here is specific to machine translation as a downstream task, unlike the proposed framework, which is general and downstream task independent. Recently, there has been more work in pretraining the decoder using BERT (Devlin et al., 2018) for neural machine translation and abstractive summarization (Edunov et al., 2019; Lample and Conneau, 2019; Song et al., 2019).

# 6   Conclusion

To answer whether unconditional language models can be conditioned to generate held-out sentences, we introduce the concept of the reparametrized sentence space for a frozen, pretrained language model, in which each sentence is represented as a point vector, which is added as a bias and optimized to reproduce that sentence during decoding. We design optimization-based forward estimation and beam-search-based backward estimation procedures, allowing us to map a sentence to and from the reparametrized space. We then introduce and use recoverability metrics that allow us to measure the effective dimension of the reparametrized space and to discover the degree to which sentences can be recovered from fixed-sized representations by the model without further training.

We observe that we can indeed condition our unconditional language models to generate held-out sentences both in and out-of-domain: our large model achieves near perfect recoverability on both in and out-of-domain sequences with $d' = 8192$ across all metrics. Furthermore, we find that recoverability increases with the dimension of the reparametrized space until it reaches the model dimension, after which, it plateaus for well-trained, sufficiently-large ($d \geq 512$) models.

These experiments reveal two properties of the sentence space of a language model. First, recoverability improves with the size and quality of the language model and is nearly perfect when the dimension of the reparametrized space equals that of the model. Second, recoverability is negatively correlated with sentence length, i.e., recoverability is more difficult for longer sentences. Our recoverability-based approach for analyzing the sentence space gives conservative estimates (upper-bounds) of the effective dimension of the space and lower-bounds for the associated recoverabilities.

We see three avenues for further work. Measuring the realtionship between regularization (encouraging the reparametrized sentence space to be of a certain form) and non-linearity would be valuable. In addition, although our framework is downstream task- and network architecture-independent, we want to compare recoverability and downstream task performance and analyze the sentence space of different architectures of language models. We also want to utilize this framework to convert encoder representations for use in a data- and memory-efficient conditional generation model.

# Acknowledgments

This work was supported by Samsung Electronics (Improving Deep Learning using Latent Structure). We gratefully acknowledge the support of NVIDIA Corporation with the donation of a Titan V GPU used at NYU for this research.

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
