[Supplementary Material · Appendix-NeurIPS.pdf]

# 7 Appendix

Figure 3: We plot the three recoverability metrics with respect to varying sentence lengths for each of our three model sizes in the 10M sentence setting. Within each plot, the curves correspond to the varying dimensions of $z$. We include error regions of $\pm\sigma$. Regardless of the divergence metric (BLEU, EM or PM), recoverability tends to improve as the size and quality of the language model improves and as the dimension of the reparametrized sentence space increases, though we see weaker overall recoverablitiy than in the better-fit 50M setting, and no cases of perfect recoverability for long sentences.

|  | $\lvert Z\rvert$ | $\overline{\overline{EM}}$ | $\sigma_{\overline{EM}}$ | $\overline{\overline{BLEU}}$ | $\sigma_{\overline{BLEU}}$ | $\overline{PM}$ | $\sigma_{PM}$ |
|---|---|---|---|---|---|---|---|
| small; 10M | 128.0 | 4.5 | 0.532 | 6.62 | 0.708 | 6.38 | 0.563 |
| small; 10M | 256.0 | 9.0 | 1.060 | 14.1 | 0.627 | 12.3 | 0.854 |
| small; 10M | 512.0 | 16.0 | 0.000 | 22.9 | 0.443 | 22.3 | 0.660 |
| small; 10M | 1024.0 | 28.5 | 1.410 | 42.2 | 0.710 | 40.1 | 0.816 |
| small; 10M | 2048.0 | 23.0 | 1.060 | 34.4 | 0.939 | 33.6 | 0.869 |
| small; 10M | 4096.0 | 34.5 | 1.600 | 46.9 | 1.160 | 45.7 | 0.961 |
| small; 10M | 8192.0 | 38.5 | 1.190 | 47.3 | 1.090 | 46.4 | 0.928 |
| small; 10M | 16384.0 | 33.0 | 1.510 | 41.0 | 1.120 | 40.1 | 0.749 |
| small; 10M | 32768.0 | 29.5 | 1.190 | 36.8 | 0.766 | 35.1 | 0.953 |
| small; 50M | 128.0 | 8.0 | 0.753 | 10.2 | 0.574 | 8.79 | 0.689 |
| small; 50M | 256.0 | 22.0 | 1.510 | 28.5 | 1.120 | 26. | 1.190 |
| small; 50M | 512.0 | 33.5 | 1.600 | 40.0 | 0.960 | 37.1 | 1.180 |
| small; 50M | 1024.0 | 64.5 | 1.190 | 71.3 | 0.821 | 69.3 | 0.801 |
| small; 50M | 2048.0 | 66.0 | 1.840 | 73.1 | 1.360 | 72.0 | 1.100 |
| small; 50M | 4096.0 | 66.0 | 1.990 | 74.0 | 1.240 | 72.2 | 1.250 |
| small; 50M | 8192.0 | 73.0 | 1.680 | 81.1 | 1.010 | 79.8 | 0.945 |
| small; 50M | 16384.0 | 70.5 | 1.920 | 76.6 | 1.270 | 74.1 | 1.100 |
| small; 50M | 32768.0 | 65.0 | 1.510 | 72.8 | 1.140 | 68.7 | 0.964 |
| medium; 10M | 128.0 | 6.5 | 0.532 | 9.26 | 0.619 | 7.56 | 0.634 |
| medium; 10M | 256.0 | 13.0 | 0.753 | 19.9 | 0.598 | 15.0 | 0.941 |
| medium; 10M | 512.0 | 28.0 | 1.060 | 35.0 | 0.993 | 30.3 | 0.975 |
| medium; 10M | 1024.0 | 39.5 | 0.922 | 45.6 | 0.623 | 42.3 | 0.200 |
| medium; 10M | 2048.0 | 71.0 | 1.060 | 76.6 | 0.660 | 75.9 | 0.874 |
| medium; 10M | 4096.0 | 67.0 | 1.840 | 75.4 | 1.150 | 73.0 | 1.090 |
| medium; 10M | 8192.0 | 71.5 | 1.600 | 79.0 | 0.813 | 77.1 | 1.050 |
| medium; 10M | 16384.0 | 66.5 | 2.060 | 74.6 | 1.030 | 72.5 | 1.010 |
| medium; 10M | 32768.0 | 67.0 | 1.680 | 76.1 | 0.812 | 71.2 | 0.920 |
| medium; 50M | 128.0 | 6.0 | 0.753 | 10.9 | 0.933 | 7.71 | 0.911 |
| medium; 50M | 256.0 | 24.5 | 0.922 | 28.0 | 0.742 | 26.6 | 0.661 |
| medium; 50M | 512.0 | 36.5 | 0.922 | 41.1 | 0.698 | 37.8 | 0.806 |
| medium; 50M | 1024.0 | 51.0 | 1.300 | 57.3 | 0.980 | 55.9 | 0.883 |
| medium; 50M | 2048.0 | 87.0 | 1.510 | 91.2 | 0.544 | 89.8 | 0.807 |
| medium; 50M | 4096.0 | 84.0 | 1.990 | 89.4 | 0.876 | 89.1 | 1.040 |
| medium; 50M | 8192.0 | 85.0 | 1.680 | 89.1 | 0.985 | 89.2 | 0.989 |
| medium; 50M | 16384.0 | 88.0 | 1.680 | 92.4 | 0.687 | 92.5 | 0.743 |
| medium; 50M | 32768.0 | 84.5 | 1.600 | 90.6 | 0.596 | 89.3 | 0.646 |
| large; 10M | 128.0 | 7.0 | 0.753 | 11.8 | 0.741 | 7.65 | 0.542 |
| large; 10M | 256.0 | 21.0 | 1.300 | 27.5 | 0.853 | 22.7 | 1.080 |
| large; 10M | 512.0 | 42.0 | 1.060 | 46.3 | 0.794 | 43.2 | 1.140 |
| large; 10M | 1024.0 | 58.0 | 1.510 | 62.1 | 1.170 | 59.9 | 1.340 |
| large; 10M | 2048.0 | 67.0 | 0.000 | 68.2 | 0.213 | 67.4 | 0.045 |
| large; 10M | 4096.0 | 95.0 | 1.300 | 97.6 | 0.577 | 97.3 | 0.529 |
| large; 10M | 8192.0 | 90.0 | 1.510 | 93.7 | 0.609 | 93.5 | 0.664 |
| large; 10M | 16384.0 | 88.5 | 1.770 | 92.4 | 0.704 | 92.6 | 0.587 |
| large; 10M | 32768.0 | 90.5 | 1.920 | 95.3 | 0.942 | 95.7 | 0.821 |
| large; 50M | 128.0 | 12.5 | 0.922 | 15.2 | 1.000 | 13.3 | 0.873 |
| large; 50M | 256.0 | 29.0 | 1.300 | 32.7 | 1.200 | 30.4 | 1.060 |
| large; 50M | 512.0 | 51.5 | 1.190 | 54.8 | 1.040 | 54.1 | 1.110 |
| large; 50M | 1024.0 | 67.5 | 0.922 | 69.5 | 0.717 | 68.4 | 0.633 |
| large; 50M | 2048.0 | 75.0 | 1.300 | 77.0 | 0.883 | 76.2 | 1.050 |
| large; 50M | 4096.0 | 99.0 | 0.753 | 99.8 | 0.204 | 99.8 | 0.189 |
| large; 50M | 8192.0 | 94.5 | 1.190 | 96.3 | 0.407 | 96.2 | 0.427 |
| large; 50M | 16384.0 | 88.5 | 1.770 | 93.8 | 0.363 | 93.8 | 0.351 |
| large; 50M | 32768.0 | 94.5 | 1.190 | 96.5 | 0.303 | 96.5 | 0.316 |

Table 2: Recoverability results when varying the dimension of the reparametrized sentence space $\lvert Z\rvert$ on exact match, BLEU, and prefix match. $\overline{\overline{EM}}$ is the sample mean over each of the 100 sentences' sample $\overline{EM}$. $\overline{EM}$ is the sample mean for a sentence over its 10 restarts. The same applies to BLEU. $\overline{PM}$ is the sample mean of prefix match medians. We also provide standard deviations.