[Reviews · NeurIPS 2019]

Reviewer 1



This paper explores an interesting question: if we are allowed certain control over the input to a pre-trained language model, can we get it to return an arbitrary sentence? The control given is a vector z associated with the sentence, which is added as a bias to the hidden state at each timestep. In forward estimation, gradient descent is used to find the optimal z to "bias" the decoder towards a given sentence. In backward estimation, a given z is decoded to find the MAP sentence it encodes (which is intractable in general, so the authors use beam search). The authors analyze the "effective dimensionality" of a sentence space given a recoverability threshold tau; that is, what's the smallest dimension such that at most a tau-fraction of sentence fail to be encoded? Interestingly, larger models seem to require lower dimensional inputs to recover sentences. The authors only check three sizes of model, but there is a convincing trend here. I want to like this paper; I'm interested in just how much can be memorized by big LMs, and I feel like this is getting at an important piece of the puzzle. But this feels like one pretty limited experiment and I'm not sure there's enough there for a strong NeurIPS paper. One technical quibble: I wonder if the results would be different if rather than a random matrix, smaller vectors were still bucketed into K buckets, the model attended over buckets, and those vectors were projected up after attention. I don't think this is equivalent to the matrix multiply version (having a softmax introduces a nonlinearity and fundamentally changes the model). There seems to be a phase transition always around 2x model dimension, which indicates to me that somehow having 4 vectors to look at really gives the model more to work with in some way. The core experiment is good and gives some interesting results. Perhaps it makes sense that in the limit, we should be able to provide a lower-dimensional indicator of a sentence and recover it, though I'm not sure what the limit of this process is, since I have no idea what the intrinsic dimensionality of language is. The major caveat in this work is whether the geometry of the z space has been meaningfully studied here. There are several limitations: (1) This assumes that all we care about are generating sentences in-domain. Gigaword is a much narrower corpus than, e.g., what BERT is trained on, and moreover much of the interest in pre-training is transfer to new settings. It seems disingenuous to pitch the model as encoding "arbitrary" sentences and then they're from the training distribution. (2) English is assumed to be a proxy for all languages. (3) More minor, but it seems like the vast majority of such models have moved to some type of generation more closely conditioned on the input (attention of various forms), so I'm not sure how practically useful this insight is for model designers. I think there's room in an 8-page paper to cover issues #1 and #2 in some depth. The authors don't necessarily need to go in this direction, but I want to see something more. Overall, I like the direction a lot, but I feel like I can't strongly endorse this as NeurIPS until there's a stronger conclusion that can be drawn. One presentation note: Section 3.2 could be written more efficiently. I was confused by the lack of detail in lines 106-130, but then this was largely recapitulated on the following page with additional details. ===================== Thanks for the nice response. I've raised my score to a 6 in light of the out-of-domain results provided; this comparison with random sentences is intriguing and tells us that what we can memorize are arbitrary *sentences* and not arbitrary strings. However, I still have an overall feeling that I'm not quite sure how to contextualize these results based on what's in the paper.

Reviewer 2



The paper investigates whether it is possible to convert the pre-trained language model into the generator that can be conditioned on the vector representation of the sentence. This conditioning can be made by adding the vector z (in one form or another depending on the dimensionality) to the hidden states of the LSTM LM. The sentence representation can be found by employing nonlinear conjugate gradient descent for maximizing the probability of the given sentence where optimization parameters are represented by the components of the vector z. Basically, the encoder of a sentence is represented by the optimization process. For sentence decoding, a beam-search is used. For sufficiently big LM almost perfect sentence recoverability from the Z space is possible. The paper explores an important direction considering recent development in unconditional text generation. However, it is not exactly clear how to apply this approach to attention-based language models (e.g. GPT), adding such discussion will clarify the limitations of the proposed method. The paper is well structured and contains all the necessary bits to understand the proposed method clearly.

Reviewer 3



The paper concerns itself with whether it is feasible to use a pretrained language model as a universal decoder. To this end, it proposes an approach to forcing a pretrained language model to output a particular target sentence, by essentially adding a bias vector to the pretrained LM's hidden state at each time step. The paper shows that for sufficiently large pretrained LMs it is possible to optimize with respect to this bias vector such that a held-out target sentence can generally be decoded (using beam search) with high accuracy. The paper is easy to follow, and the idea of a universal decoder is compelling (and likely to be on the minds of many NLP people), and so I think the results presented in this paper will have an impact. At the same time, the question of whether it will actually be practical to have a universal decoder remains unanswered, since (as the authors partially note) it is unclear whether a pretrained LM can generate text that is sufficiently different from that on which it was trained, it is unclear whether an encoder can mimic the optimization with respect to the z's (though it seems reasonable to be optimistic about this), and it is not clear whether the finite capacity of the z vectors will end up being a practical issue, especially for generating longer sentences. The experiments are largely convincing. However, although it isn't completely clear, it sounds as though the beam search is carried out assuming the true length T, which may lead to an overly optimistic numbers. Similarly, it would be good to establish whether larger beams ever decrease performance (as they often do), which might be another reason for caution. Update after response: thanks for the out-of-domain results and the beam search experiments. These results are certainly encouraging, and I continue to recommend acceptance.

[Author Response · NeurIPS 2019]

**Author Response:**  We thank all our reviewers for their careful evaluation and numerous suggestions.

**Reviewer 1**  We agree that evaluating on just in-domain sentences is a limitation. As a result, we will add recoverability results on a subset of 50 sentences from the validation set of the IWSLT16 En-De translation dataset to the paper. This corpus is composed of TED talk transcripts, a very different style from the news text our language models were trained on. Further, we randomly sample another 50 sentences of varying lengths where each token is sampled randomly with equal probability with replacement from the vocabulary structure to evaluate whether $z$ memorizes the sentence independent of the language model. We evaluate this across all our model sizes on the language models that have seen 50M sentences. For space constraints we provide two representative graphs over the IWSLT16 data (large and medium models) and one representative graph over the Random token sampling data (large model).

We are able to recover out-of-domain sentences well and thus seems to account for the domain shift from news to TED transcripts well. Recoverability estimates are nearly perfect for both the large and medium models. We observe that on the random data we are able to memorize sentences up to a length of approximately 25 with large $z$, but this drastically degrades after that–indicating that memorization does not fully explain results on Gigaword and IWSLT. We will add these to the paper. Our results and conclusions hold only for English, and this is a limitation of the work. We are also interested in your proposition about performing attention over smaller $z$'s. It isn't equivalent to matrix multiplication and could accelerate learning, but we have to leave these for future work.

**Reviewer 2**  We missed the Mu et al 2017 paper on representing sentences with low-rank subspaces, which we have now added to the related work section. Our definition of the reparametrized sentence space is language-model agnostic and thus applicable to GPT-2. However, since GPT-2 uses a Transformer (not an RNN) decoder, we must change how we reparametrize the sentence space and figure out where to integrate $z$. One possible choice is to use $z$ as an additional bias in each feed-forward layer of every word in a given sentence. To address the impact of optimization strategy, we reran a some of our best performing settings with Adam with a learning rate of 1e-4 and default parameters. We find that Adam results in $z$'s that do not exceed 1.0 BLEU (Table 1). We will add this to the paper.

**Reviewer 3**  We don't explicitly assume a true length when decoding with beam search—we stop when an end of token or 100 tokens is reached. Some of the degenerate candidate recoveries repeat a specific token until the 100 token limit is reached. Larger beam sizes do not significantly change performance, see Table 2. We provide a partial table below and will add a couple of sentences to the paper stating this and clarifying the previous point.

Table 1: Optimizer results on English Gigaword

| Model | $|Z|$ | BLEU | |
| --- | --- | --- | --- |
| | | ConjGrad | Adam |
| SMALL; 50M | 8192 | 81.1 | 0.031 |
| MEDIUM; 50M | 16384 | 92.4 | 0.234 |
| LARGE; 50M | 4096 | 99.8 | 0.037 |

Table 2: Beam width results on English Gigaword

| Model | $|Z|$ | BLEU | | |
| --- | --- | --- | --- | --- |
| | | Width=5 | Width=10 | Width=20 |
| SMALL; 50M | 512 | 40.0 | 40.3 | 40.5 |
| SMALL; 50M | 8192 | 81.1 | 79.8 | 79.6 |
| MEDIUM; 50M | 512 | 41.1 | 41.1 | 42.3 |
| MEDIUM; 50M | 16384 | 92.4 | 91.9 | 89.8 |
| LARGE; 50M | 512 | 54.8 | 54.1 | 53.8 |
| LARGE; 50M | 4096 | 99.8 | 99.8 | 99.5 |

[Meta-Review · NeurIPS 2019]

While we now understand how we can pretrain text encoders or non-conditional language models, the important open question is figuring out a method for pretraining (or using pretrained) decoders in seq2seq models. While several proposals have been made, neither was particularly successful. This paper does not deliver this either but it answers a very natural question anyone working on this problem would like to ask -- can we even steer a pretrained language models so that it generates a given sequence? I (as well as) reviewers found the paper (very) interesting: the study is well executed, well written and provides new insights into the properties and limitations of pretrained language models. There is consensus that the paper should be accepted.